# IQCELL: A platform for predicting the effect of gene perturbations on developmental trajectories using single-cell RNA-seq data

Tiam Heydari[1,2], Matthew A. Langley[3¤a], Cynthia L. Fisher[1,2], Daniel Aguilar-Hidalgo[1,2], Shreya Shukla[3,4], Ayako Yachie-Kinoshita[3¤b], Michael Hughes[5], Kelly M. McNagny[1,5], Peter W. Zandstra[1,2]*

1 School of Biomedical Engineering, University of British Columbia, Vancouver, British Columbia, Canada,
2 Michael Smith Laboratories, University of British Columbia, Vancouver, British Columbia, Canada,
3 Institute of Biomaterials and Biomedical Engineering, University of Toronto, Toronto, ON, Canada, 4 Notch Therapeutics, Vancouver, British Columbia, Canada, 5 Department of Medical Genetics, University of British Columbia, Vancouver, BC, Canada

¤a Current address: Division of Biology and Bioengineering, California Institute of Technology, Pasadena, United States of America
¤b Current address: The Systems Biology Institute, Shinagawa, Tokyo, Japan
* peter.zandstra@ubc.ca

**Data Availability Statement:** The source code of IQCELL python package generated during this study and example notebooks of IQCELL's

## Abstract

The increasing availability of single-cell RNA-sequencing (scRNA-seq) data from various developmental systems provides the opportunity to infer gene regulatory networks (GRNs) directly from data. Herein we describe IQCELL, a platform to infer, simulate, and study executable logical GRNs directly from scRNA-seq data. Such executable GRNs allow simulation of fundamental hypotheses governing developmental programs and help accelerate the design of strategies to control stem cell fate. We first describe the architecture of IQCELL. Next, we apply IQCELL to scRNA-seq datasets from early mouse T-cell and red blood cell development, and show that the platform can infer overall over 74% of causal gene interactions previously reported from decades of research. We will also show that dynamic simulations of the generated GRN qualitatively recapitulate the effects of known gene perturbations. Finally, we implement an IQCELL gene selection pipeline that allows us to identify candidate genes, without prior knowledge. We demonstrate that GRN simulations based on the inferred set yield results similar to the original curated lists. In summary, the IQCELL platform offers a versatile tool to infer, simulate, and study executable GRNs in dynamic biological systems.

## Author summary

Executable GRNs provide an important strategy to model complex intracellular dynamics in development and disease. Here we introduce IQCELL, a platform to infer, simulate, and study executable logical GRNs directly from single cell sequencing data. IQCELL is an integrative platform that includes modules for gene selection, building logical GRNs, and simulating developmental trajectories under normal and perturbed conditions. We

implementation are available on Gitlab: (https://
gitlab.com/stemcellbioengineering/iqcell). The raw
sequencing data generated in this study have been
submitted to GEO under the accession number
GSE196972.

**Funding:** TH, MAL, AYK, CLF, DAH, SS, and PWZ
were supported by the Canadian Institutes of
Health Research (CIHR), Foundation Grant FRN
154283, and the Natural Sciences and Engineering
Research Council of Canada (NSERC), Discovery
Grant RGPIN-2020-06496, to PWZ. PWZ is the
Canada Research Chair in Stem Cell Bioengineering
(https://www.chairs-chaires.gc.ca). The funders
had no role in study design, data collection and
analysis, decision to publish, or preparation of the
manuscript.

**Competing interests:** The authors have declared
that no competing interests exist.

demonstrate the utility of IQCELL by reconstructing GRNs for early mouse T-cell and red
blood cell development. We show that IQCELL can "automatically" infer the vast majority
of gene interactions previously reported from decades of experimental research. IQCELL
also provides users with a platform to simulate the developmental trajectories of cells. We
show that dynamic simulations of the inferred GRNs resemble experimentally observed
gene expression dynamics and capture the effects of genetic perturbation studies. IQCELL
offers a versatile tool to infer and simulate GRNs in dynamic biological systems.

## Introduction

Stem cell fate decisions are made via dense arrays of interacting transcription factors (TFs)
forming gene regulatory networks (GRNs) [1]. Information gleaned from GRNs during stem
cell differentiation can lead to more effective design-based cell cultures, applicable to cell thera-
pies [2,3]. As a prominent example, the effect of TFs on GRNs has been widely utilized in the
reprogramming of embryonic and adult somatic cell GRNs for the establishment of a pluripo-
tent state via induction of driver TFs [4]. Stem cell reprogramming and differentiation can be
modeled as executable and logical (Boolean) GRNs undergoing state transition [5–7]. Execut-
able GRNs provide information about both the topology and the regulatory rules of gene inter-
actions that can be simulated as time-evolving (dynamical) systems. However, deriving
informative, executable, and predictive GRNs for stem cell differentiation has proven to be a
challenging task. Specifically, developing executable GRNs by piecing together evidence from
gene perturbation experiments has shown to be an effective strategy [6] but is extremely time-
consuming, labor intensive, and expensive. In a notable advancement, automated formal rea-
soning successfully identified a set of minimal GRNs underlying naive pluripotency in mice.
Gene expression observations across multiple culture conditions were used to logically con-
strain possible GRN configurations, and the resulting set was able to accurately predict the out-
come of 70% of new experiments [8,9]. Yet, these methods are not based on high-throughput
data.

More recently, the emergence of single-cell profiling technologies has provided an unprece-
dented archive of information regarding cells undergoing fate determination and maturation.
Deriving more accurate GRNs based on single-cell data is at the center of many recent efforts
[10–12]. Formal reasoning has been used to infer executable GRNs directly from high
throughput single-cell quantitative PCR (sc qPCR) data [13,14]. However, using single-cell
RNA-sequencing (scRNA-seq) data has many advantages in terms of coverage, availability,
flexibility in gene selection, and accuracy in clustering and pseudo-time inference compared
to sc qPCR. These benefits exist alongside the disadvantage of dropout effects and low sensitiv-
ity in profiling TFs. Despite the availability of this data resource, this emerging field is still
missing an integrated platform to infer, study, and simulate executable GRNs directly from
scRNA-seq.

Herein we report an effective platform (IQCELL), implemented as a Python software pack-
age for reconstructing GRNs directly from scRNA-seq data. Our strategy includes steps for
gene selection, correcting dropout effects, building logical GRNs directly from pseudo-time
with respect to interaction hierarchy and mutual information between gene pairs, and simulat-
ing developmental trajectories under normal and perturbed conditions. We demonstrate the
utility of IQCELL by reconstructing GRNs for early mouse T-cell and red blood cell develop-
ment, well characterized mammalian developmental systems [15,16], using published scRNA-
seq datasets [17,18]. Our resulting GRNs based on a curated set of genes, recover over 74% of

experimentally validated causal gene-gene interactions spanning years of research. Dynamic simulations of the inferred GRN resemble experimentally observed gene expression dynamics and capture the effects of knocking out or forcibly expressing various genes. Next, we use IQCELL's TF selection pipeline to automatically infer a set of genes to construct a GRN. We show that the pipeline selects many previously known TFs that are important for T-cell and red blood cell development, without prior knowledge. Finally, we show that perturbation outcomes from the unbiased GRN are highly concordant with the perturbation outcomes from the GRN made based on the curated gene lists. Our platform is applicable to scRNA-seq data derived from dynamic developmental systems and should serve as a useful resource for many applications.

## Results

### An integrated platform for predicting the qualitative effects of gene perturbations on developmental trajectories (IQCELL)

IQCELL infers logical regulatory networks directly from existing information in the scRNA-seq data of cells during development and uses these regulatory networks to simulate and predict the behavior of developing cells under perturbed conditions (**Fig 1**). IQCELL works with quality controlled and pre-processed scRNA-seq gene expression data [19,20]. The second input of the IQCELL platform is the inferred pseudo-time ordering of cells based on scRNA-seq data (**Fig 1**). The temporal dynamics of genes helps with the inference of causal gene interactions. Pseudo-time ordering of cells using scRNA-seq data has been shown to be informative for capturing temporal and developmental dynamics [21,22].

IQCELL has a customized gene selection pipeline based on the pySCENIC platform [23]. In this pipeline (**S1A Fig**), the TFs are selected based on the highly active and dynamic regulons (see Methods). Since gene dropout is common among many scRNA-seq datasets, particularly for TFs with low mRNA copy numbers, IQCELL employs a recently developed graph-based

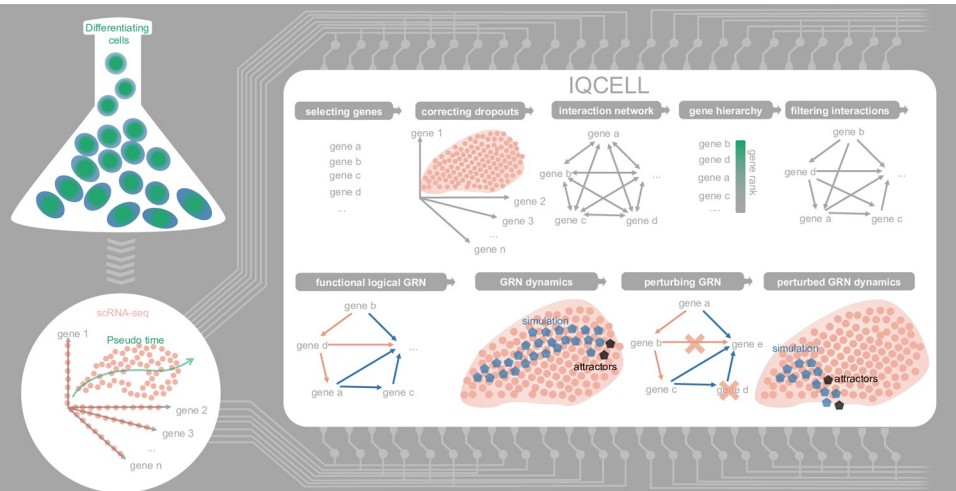

**Fig 1. Overview of IQCELL.** IQCELL infers logical GRNs directly from sc-RNA seq data and allows the simulation and analysis of *in silico* developmental trajectories in normal and perturbed conditions. The typical inputs of IQCELL are sc-RNA seq expression data along with the pseudo-time ordering of the cells. After correction of dropout effects and gene selection steps, gene-gene interactions are calculated and weighted based on mutual information. Binarized gene expression values are used to constrain possible gene-gene interactions and obtain a functional GRN for the data. IQCELL can be used to analyze the GRN and simulate possible developmental trajectories under normal and perturbed conditions.

algorithm (MAGIC) to recover gene expression [24], (**S1B Fig**). Next, IQCELL generates a set of possible interactions between genes. Information-based metrics such as mutual information are well suited for quantifying relationships between genes [25]. IQCELL scores gene-gene interactions according to the mutual information between gene pairs [26] and assigns a regulatory sign (activation or repression) to each interaction based on the significance and sign of their correlation (**S1C Fig**). These steps result in a dense weighted network of gene-gene interactions that needs to be filtered into a functional GRN. In a functional GRN, interactions are not necessarily biophysically direct but capture the consequence of regulatory relations.

To reduce the number of possible gene interactions, IQCELL forms a gene interaction hierarchy in which higher ranked genes influence lower ranked ones. To form this hierarchy, IQCELL binarizes the gene expression counts by clustering them into expressed and non-expressed states [27]. The binarization process divides the pseudo-time axis into regions with compact and sparse expression densities for the genes, reflecting the pseudo-time domains where a gene is expressed at a higher or lower level (**S1D Fig**). Next, the platform identifies the transition points between expression regions for all genes and uses the order of transitions to form a gene interaction hierarchy, with highly ranked genes (with earlier transition points) having greater potential to influence those downstream. This acts as an additional filter on gene-gene interactions along with the mutual information (**S1E Fig**). The resulting directional network serves as the foundation for inferring executable GRNs.

To obtain an executable GRN model, IQCELL models interactions between genes as Boolean logic functions [7]. IQCELL uses a satisfiability modulo theory engine (Z3) (**S1F Fig**), [28] to identify logic functions that are compatible with the pseudo-time dynamics of binarized gene expression states [13]. Finally, it selects the GRN with the highest average mutual information as the most probable constrained model. The result is a functional and executable GRN that optimally fits the input scRNA-seq data. IQCELL has built-in capabilities to simulate GRN dynamics via random asynchronous Boolean simulation [7] and compare the results with experimental data under normal and perturbed conditions. In summary, IQCELL processes scRNA-seq data inputs to infer an executable logical GRN that best fits the data.

## IQCELL sorts genes based on transition points and places them in a biologically relevant order

To assess the functional capabilities of IQCELL, we evaluated its performance using a well characterized mammalian developmental system: mouse early T-cell development [29,30]. The T-cell developmental program takes place in the thymus, where pre-thymic progenitors differentiated within the bone marrow progress toward T lineage commitment, involving a dense network of genes [31]. Sustained exposure to Notch signaling drives early thymic progenitors (ETP) to the *CD4/8* double-negative 2a (DN2A) and 2b (DN2B) stages, where upregulation of T-cell lineage-specific genes and progressive loss of potential for other blood cell fates occurs. Once committed to the T-cell fate, the double-negative 3 (DN3) T-cell progenitors begin recombining the β-chain of the pre-T-cell receptor (TCR). Cells are selected for functional β-chain rearrangements through pre-TCR signaling and proceed toward *CD4/8* double positive state (DP) (**Fig 2A**). We used a publicly available scRNA-seq dataset [18], where the authors used fluorescence-activated cell sorting to capture mouse thymocytes at ETP-DN2 and DN3 stages based on cell surface markers. After processing the data in a manner consistent with the original publication (**S4 Fig** and **S2 Table**), the gene expression profiles and the pseudo-time orders were used as inputs to IQCELL.

After expression recovery, selecting a curated list of genes that are known to be important during early T-cell development (**S1 Table** and **Fig 2B**, see METHODS), and finding possible

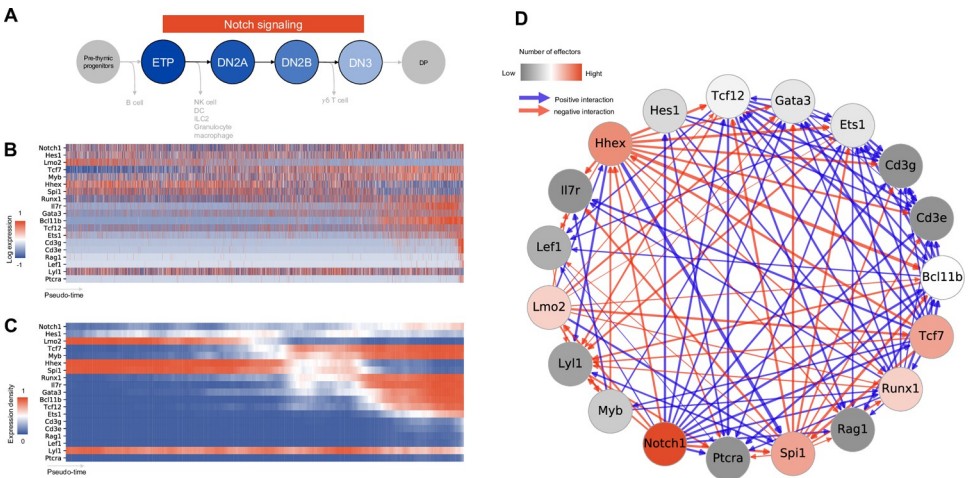

**Fig 2. IQCELL initial processing of early T-cell development sc-RNA seq data.** (A) Summary of the scope of the sc-RNA seq data used as an input to IQCELL [18]. ETPs originated from pre-thymic progenitors progress toward DN2A, DN2B (coincides with upregulation of Bcl11b and lineage commitment), DN3 stages and eventually lead to DP cells (not covered here). (B) Log transformed expression matrix for selected genes from sc-RNA data along the pseudo-time axis. Gene expression is corrected for dropout effects using MAGIC [24]. Red indicates high expression, blue indicates low expression. (C) Smoothed binarized gene expression matrix (expression density). Gene expression values were binarized by clustering, averaged over a pseudo-time window, then sorted based on transition points from early to late. Red indicates high expression, blue indicates low expression. (D) The set of all possible gene-gene interactions, filtered by interaction hierarchy and mutual information. Positive and negative interactions are represented by blue and red edges, respectively. Edge width represents the relative amount of mutual information of the interaction.

gene-gene interactions, IQCELL binarized gene expression values and calculated the expression density over pseudo-time (**Fig 2C**) (see Methods). Sorting the genes based on their transition points placed *Notch1* and *Hes1* at the top of the gene interaction hierarchy (since their expression level stayed relatively high consistently) followed by *Lmo2*, *Tcf7*, *Myb*, and *Runx1* which agrees with their position in the regulatory hierarchy during T-cell lineage establishment [30]; whereas DN3 associated genes such as *Cd3e*, *Lef1*, and *Ptcra* [32,33] appeared at the bottom of the hierarchy (**Fig 2C**).

Next, IQCELL used this order of genes (**Fig 2C**) as a hierarchical filter of possible interactions, with the genes at the top having the most regulatory potential in terms of number of genes they can regulate. The combination of the regulatory potential of individual genes, the mutual information between gene pairs, and interaction signs, led to a directed gene interaction network comprising the set of possible interactions (**Fig 2D**). This network then constitutes a foundation for further constraints and analyses at next steps.

## IQCELL is highly predictive for functional regulatory interactions

Following our *in silico* analysis, we compared predictions from our initial inferred interaction network to validated regulatory interactions in mouse T-cell development. The initial interaction network (**Fig 2D**) was simplified with additional constraints. These constraints enforce the gene interactions to follow the expression patterns throughout the pseudo-time axis (**Fig 2B**) when executed as a logical network. This resulted in a set of possible update logical rules for each gene, selecting the most probable interactions as scored by mutual information leading to a provisional executable GRN for early T-cell development (**Fig 3A** and **S1 Table**).

To benchmark this GRN, we compared our predicted directional interactions with a recent comprehensive GRN model of mouse T-cell development based on experimentally validated gene interactions [16]. This network consists of 38 reported interactions between the genes of

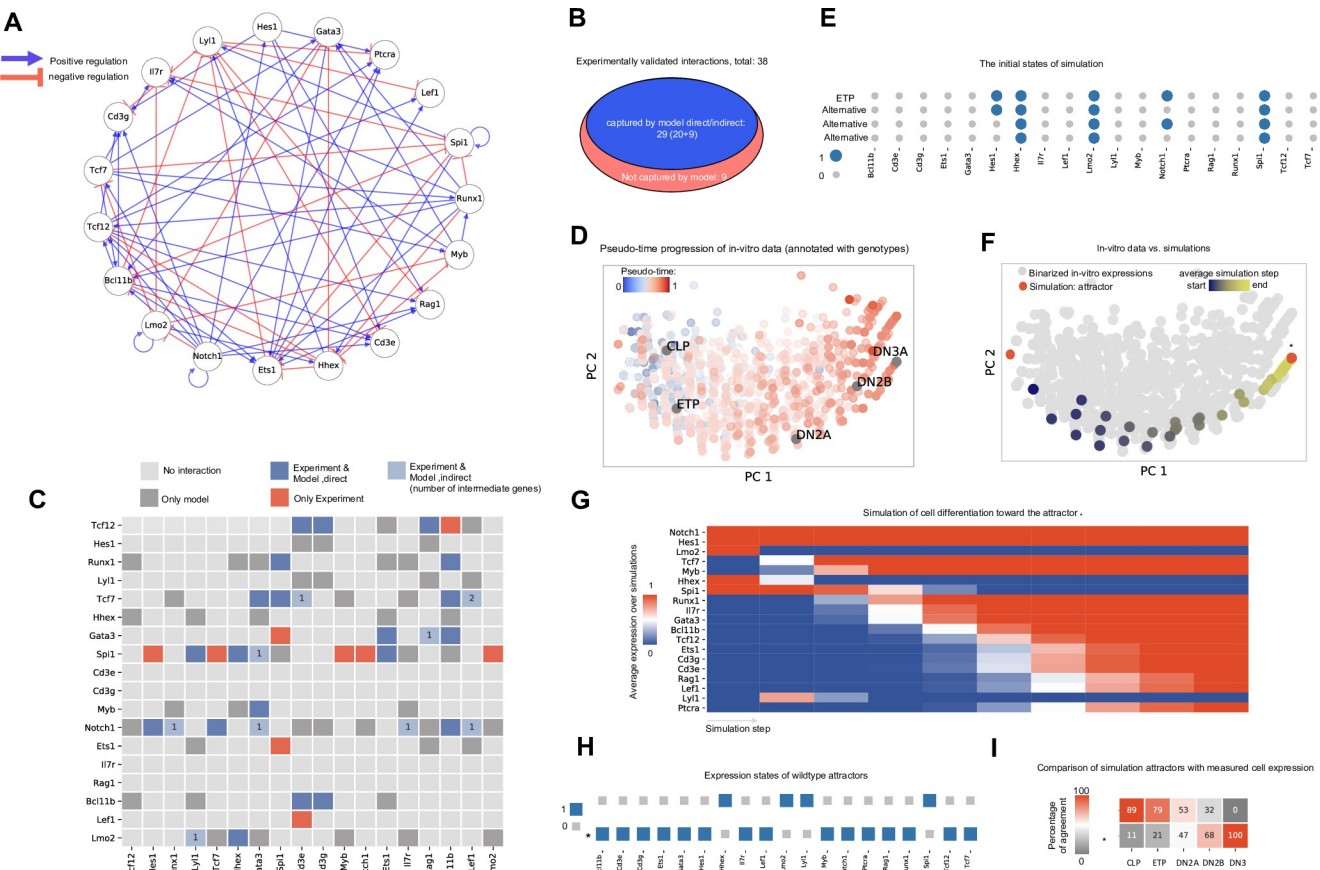

**Fig 3. The provisional GRN for mouse early T-cell development inferred by IQCELL captures essential gene interactions and accurately simulates T-cell developmental trajectories.** (A) The provisional GRN for early mouse T-cell development. The GRN is obtained by constraining the possible interactions to both follow the *in vitro* data progression when executed as a logical network and maximize mutual information between gene pairs. Positive and negative interactions are represented by blue and red edges, respectively. (B) Out of 38 experimentally reported gene interactions of early mouse T-cell development [16], 29 of them are captured by the functional GRN model proposed by IQCELL. (C) Detailed representation of the proposed interactions by IQCELL and experimentally reported ones. Rows and columns represent regulators and effector genes, respectively. Blue indicates that the interaction is captured by the model directly (dark blue) or indirectly (light blue); in the latter case, the numbers indicate the number of intermediate genes. Dark gray indicates that the interaction is only proposed by IQCELL. The red color indicates the experimentally validated interaction is not present in the model. Light gray cells indicate no interaction. Genes downstream of *Spi1* comprise 50% of the experimentally-reported interactions not captured by IQCELL. (D) The PCA plot of the binarized scRNA-seq data color-coded with the pseudo-time values attributed to each cell. The binarization is performed by clustering the scRNA-seq expressions into expressed or not expressed levels. On top of that, the binarized expressions of CLP, ETP, DN2A, DN2B, and DN3A cells have been calculated from the Immgen microarray data [38] and overlaid on RNA-seq data. (E) The four initial states that have been used in simulations. Three variations of the state representing ETP are due to the noisy expressions of Notch1 and Hes1 genes in recovered sc-RNA seq data with early pseudo-time. Genes that are expressed (1) and not expressed (0) are represented with blue and grey circles, respectively. (F) The PCA plot of the simulated developmental trajectories are overlaid on the binarized scRNA-seq. The two detected attractors are colored red, and the attractor that matches the DN3A state is marked by star (*). The simulated data is color coded by the value of average simulation step (average distance to the attractor of simulation). (G) Average gene expression at each simulation step. All simulations started from the same initial condition (ETP) and move toward the same attractor (*). (H) Expression states of the GRN model steady state attractors. Genes that are expressed (1) and not expressed (0) are represented with blue and grey squares, respectively. (I) Percentage of similarity between the two attractors (vertical axis) and binarized microarray expression profiles of CLP, ETP, DN2A, DN2B, and DN3A cells (horizontal axis) [38]. The average agreement between two random states is 50%.

interest, of which 29 (over 75%) are de novo captured directly by our simulated functional regulatory network (**Fig 3B** and **3C**). For example, it is well known that *Bcl11b*, the gene that marks T lineage commitment, is activated by Notch signaling, *Gata3*, *Tcf7*, and *Runx1* [34]. Our model predicted four activators for *Bcl11b*: *Notch1* (as a Notch signaling mediator [35] and target gene [36] functionally represents the presence of Notch signaling), *Gata3*, *Tcf7*, and *Runx1*. The presence of *Runx1* in our model is a notable result. *Runx1* is present in developing

cells, however it only gains access to the *Bcl11b* locus after chromatin restructuring during the DN2A stage [37]. Notably, more than half of the interactions that were not captured by our model are related to the *Spi1* gene which is a T-cell lineage suppressor [30].

To test the performance of IQCELL with another source of data on T-cell development under different conditions, we performed scRNA-seq analysis of *in vitro* differentiation of fetal liver hematopoietic progenitor cells toward the T-cell lineage. Application of IQCELL to this second scRNA-seq data set provided further validation of its ability to predict gene-gene interactions (**S3 Fig**).

One advantage of logical GRN models is that they can not only provide information about gene interactions, but can also be simulated to predict how the system evolves in time. To demonstrate this capability, we simulated our inferred logical GRN model and compared its output to experimental observations of mouse T-cell development. The scRNA-seq expression data [18] was binarized by grouping the gene expression count into on and off states. This data was then used in principle component analysis (**Fig 3D**) and the simulated trajectories overlaid on top of the binarized scRNA-seq gene expression data (**Fig 3F**). As the initial states of the simulations (representing the starting expression state of simulations), we used the binarized representation of cells at the beginning of the pseudo-time trajectory. These cells resemble the known expression state of ETP cells [30]. However, given the noisy expression of *Notch1* and *Hes1* at the earlier pseudo-time points (**Fig 2C**), we considered the expression states of these two genes to be random which results in four distinct initial states in total (**Fig 3E**). Two steady states have been obtained for the given initial cell states (**Fig 3F**), with one of them matching the DN3A cell profile (noted by star * in Fig 3). The simulated gene expression dynamics from ETP state towards this steady state shows a similar trajectory compared to the one observed from scRNA-seq data (**Fig 3G,** compare with **Fig 2C**). The other steady state shares similarities with common lymphoid progenitors (CLP) and ETP cells (**Fig 3H** and **3I**). Overall, this analysis demonstrates that our GRN model is informative about both gene interactions and the behavior of genes at the system level. Such a model has the potential to predict the effect on gene perturbations at the system level as well.

## IQCELL predicts the effect of gene perturbations on developmental trajectories

Next, we tested the effect of gene perturbations on simulated developmental trajectories (**Fig 4A**). Specifically, we tested the effect of gene perturbations known to result in halting or promoting T-cell development during the ETP-DN3 stages (reviewed in [30]).

*Notch1*, a cell surface receptor that mediates Notch signaling, is known to play an essential role in early T-cell development. *Notch1* deficiency leads to blocked T-cell development and accumulation of other hematopoietic lineages [35]. Using our inferred executable network, we simulated the developmental trajectory of ETP cells in the absence of *Notch1*. Simulations predicted the presence of two possible steady state attractors, localized near the earlier section of the pseudo-time domain (**Fig 4B**). Comparing the expression states of the simulation attractors (**Fig 4C**) with the binarized expression of known cell states extracted from microarray data [38], we found that the attractor states are more similar to ETP or CLP, and none show significant similarities to later stages of T-cell development (**Fig 4D**). This agrees with previous reports [35] that lack of *Notch1* blocks T-cell development.

*Tcf7* is a crucial TF for T-cell specification and differentiation that is upregulated by Notch signaling. Lack of *Tcf7* results in premature arrest of T-cell development before the DN2 stage [39]. Our model predicts a single attractor state in the absence of *Tcf7* (**Fig 4B**). This attractor precedes the DN2 stage and does not express *Gata3*, *Bcl11b*, *Ets1*, *Cd3e*, or *Cd3g* (**Fig 4C**), in

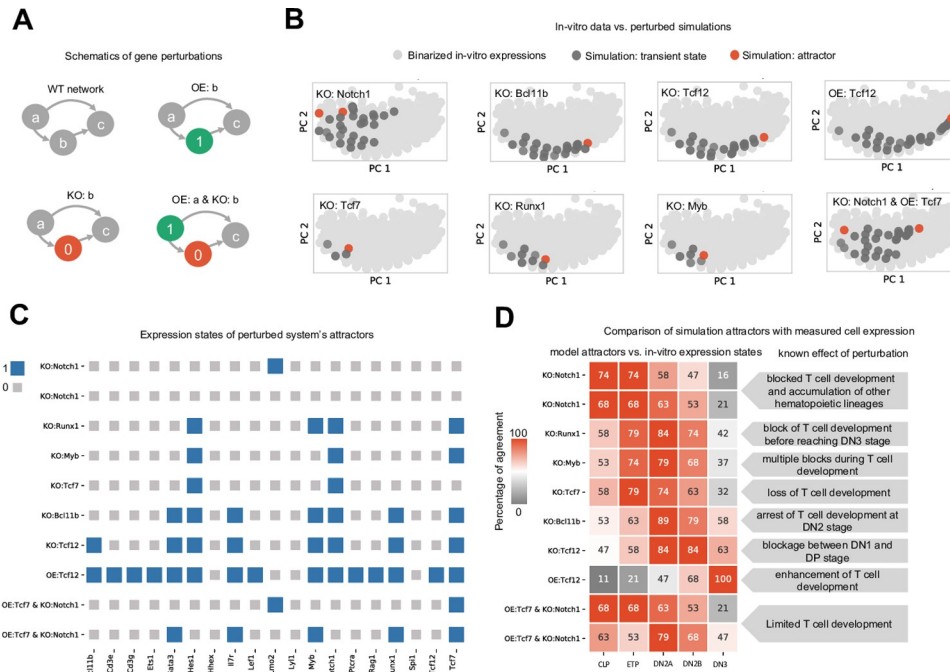

**Fig 4. Testing the known effect of eight gene perturbations on *in silico* developmental trajectories.** (A) Schematic of performed gene perturbations. In OE, the gene is always expressed (represented with 1) and in KO the gene is always silent (represented with 0). (B) PCA plot of the simulated developmental trajectories under perturbed conditions are overlaid on the binarized scRNA-seq. The perturbations include KO of *Notch1*, KO of *Tcf7*, KO of *Bcl11b*, KO of *Runx1*, KO of *Tcf12*, KO of *Myb*, OE of *Tcf12* and the double perturbation, OE of *Tcf7* and KO of *Notch1* at the same time. (C) Expression states of the model attractors under perturbations. Genes that are expressed (1) and not expressed (0) are represented with blue and grey squares, respectively. (D) Percentage of similarity between the model attractors under perturbations (vertical axis) and the binarized expressions of CLP, ETP, DN2A, DN2B, and DN3A cells (horizontal axis) [38] (left). Description of known effect of the gene perturbation on T-cell development (right).

agreement with experimentally reported analysis of *Tcf7*-/- lymphoid-primed multipotent progenitors cultured *in vitro* (on OP9-DL4) at day 4 [39]. The simulated *Tcf7* knockout (KO) steady state attractor shows more similarity to microarray profiles for ETP cells than either DN2 and DN3 cells (**Fig 4D**), which is in agreement with previous reports [39].

Next, we investigated the effect of simulated KO of *Bcl11b*, a crucial gene for T-cell commitment [40]. It has been shown experimentally that *Bcl11b* deficient cells cannot proceed beyond the DN2 stage [41]. Our *in silico* results predict one attractor in the absence of *Bcl11b* (**Fig 4B**). This attractor resembles the DN2A cell state (**Fig 4D**) which recapitulates the aforementioned experimental result of *Bcl11b* KO [41].

We also simulated the effects of perturbing *Runx1*, *Tcf12* and *Myb*. KO of *Runx1* stops T-cell development before the DN3 stage [42]; KO of *Tcf12* results in developmental blockage before DP stage [43], whereas *Tcf12* overexpression (OE) enhances T-cell development[44]; KO of *Myb* causes multiple blocks during T-cell development [45]. We have simulated these gene perturbations, and found qualitative agreement with the experimentally reported results (**Fig 4B, 4C** and **4D**).

Next, we tested our model for a simultaneous perturbation of two genes. Interestingly, while the absence of Notch signaling results in loss of T-cell development, forced expression of *Tcf7* can partially rescue T-cell development in the absence of Notch [39]. To test our model against this observation, we simulated shutting off the expression of *Notch1* and forcing

expression of *Tcf7* to the 'on' state simultaneously. This resulted in two attractors (**Fig 4B**), one of them localized at earlier stages, and one of them close to the DN2 stage, which reflects the limited but not complete development of T-cells when compared with the KO of *Notch1* (**Fig 4C** and **4D**).

In addition to these perturbations, we have performed full systematic GRN perturbations for one and two gene perturbations (**S2A Fig** and **S3** and **S4 Tables**) and sensitivity analysis of gene interactions (**S2B Fig** and **S3** and **S4 Tables**). Taken together, we showed that our model can predict the effect of single and double gene perturbations on the developmental trajectory of early T-cell development.

## An automated TF selection module can replace the need for a curated gene list

In many biological systems, validated lists of curated relevant genes are not known *a priori*. To enable the unbiased construction of predictive executable GRN, we developed a customized pipeline that automatically selects a set of TFs directly from scRNA-seq data (**Fig 5A**). This pipeline starts by identifying a set of 2000 highly variable genes (HVGs) from the data. Next, it leverages the pySCENIC platform [23] to select more than 200 TFs that are components of core regulons during the differentiation process. As a final step, the module selects a subset of these genes based on their gene dynamics score (GDS), (**S5A Fig**), (see METHODS for more details).

We applied this pipeline to the previously utilized scRNA-seq dataset [18] and selected the top 14 genes based on GDS. Our analysis automatically captured many biologically important

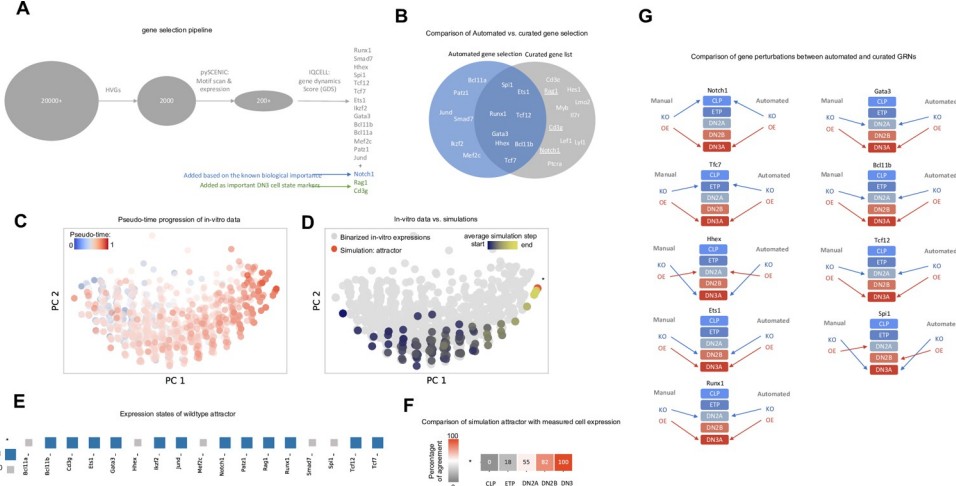

**Fig 5. Constructing mouse early T-cell GRN based on automated TF selection pipeline.** (A) Overview of TF selection procedure. After selecting HVGs, IQCELL uses pySCENIC to select active regulons (TFs and their effectors), and finally IQCELL uses GDS to rank and select TFs for the final list. We have added Notch1 for its known biological importance and *Rag1* and *Cd3g* as biological markers of DN3 stage to the list. (B) Comparison of automated vs curated TF selection show that the TF selection pipeline captures 8 of 14 genes in the curated gene list. (C) The PCA plot of the binarized scRNA-seq data color-coded with the pseudo-time values attributed to each cell. The binarization is performed by clustering the scRNA-seq expressions into expressed or not expressed levels. (D) The PCA plot of the simulated developmental trajectories are overlaid on the binarized scRNA-seq. The detected attractor is colored red and marked by star (*). The simulated data is color coded by the value of average simulation step (average distance to the attractor of simulation). (E) Expression states of the GRN model steady state attractor. Genes that are expressed (1) and not expressed (0) are represented with blue and grey squares, respectively. (F) Percentage of similarity between the attractor (vertical axis) and binarized microarray expression profiles of CLP, ETP, DN2A, DN2B, and DN3A cells (horizontal axis) [38]. The average agreement between two random states is 50%. (G) Comparison of gene perturbations between automated and curated GRNs show 17 matches of the predicted sates out of 18 perturbations.

TFs involved in mouse T-cell development including *Tcf7*, *Gata3*, *Runx1*, *Bcl11b*, and *Tcf12*. As discussed before, a known factor in T-cell development is Notch signaling (represented by *Notch1*), we added this gene along with two DN3 stage markers (*Cd3g* and *Rag1*) to the gene set to create the final gene list (**Fig 5A**). Comparing the automated TF selection with the curated gene list shows that 8 out of 14 TFs that are selected via the pipeline are in common with the curated gene list (**Fig 5B**).

Next, we followed IQCELL's GRN inference pipeline to obtain a functional GRN for the system (**S5A–S5E Fig**). The dynamic simulation of the GRN matches the scRNA-seq data (**Fig 5C and 5D**) and the attractor state matches the DN3 stage (**Fig 5E and 5F**).

Next, we performed systematic KO and OE simulations on genes of the GRN (**S5F and S5G Fig**), including the 8 TFs and *Notch1* which are in common with the curated gene list. Comparing the predictions of two GRNs on these 9 genes showed that the resulting state of 17 out of 18 perturbations agreed. The only difference is the outcome of OE of *Spi1*gene (DN2A vs DN2B).

To conclude we have shown that the IQCELL TFs selection pipeline is able to capture biologically important genes and the predicted effects of gene perturbations are robust against the selected gene set.

## Testing IQCELL performance in red blood cell development

To test IQCELL platform on an additional data set, we evaluated its performance in the context of erythropoiesis, the development of red blood cells. Erythroid progenitors (ErP) arise from megakaryocyte/erythroid progenitors (MEPs) via a lineage bifurcation process [46] (**Fig 6A**). IQCELL performance in this context was evaluated using a publicly available dataset of mouse hematopoietic stem and progenitor cell differentiation [17]. This dataset is an atlas of single-cell HSPC expression and the differentiation of these cells into multiple lineages. Building on a recent analysis of this dataset [47], we isolated the MEPs and ErP subclusters from the dataset, and constructed a pseudo-time ordering of the cells connecting MEPs to ErPs. Initially we used a curated list of erythropoiesis—associated genes and a well characterized interaction network [15] including *Gata1*, *Gata2*, *Klf1*, *Fli1*, *Spi1*, *Zfpm1*, and *Zbtb7l* (**Fig 6B**).

Following the previously described IQCELL GRN inference pipeline we calculated the expression density over pseudo-time after the expression recovery step for the curated gene list (**S6A Fig**) and constructed the initial interaction network (**S6B Fig**). This resulted in an executable GRN model (**Fig 6C**), which includes 11 interactions among 7 genes. Notably, IQCELL correctly predicted the target genes of *Gata1* including *Zfpm1* (known as *FOG-1*), *Spi1*, *Klf1* and *Gata2* (the detailed interaction rules can be found in **S5 Table**). Comparing the known gene-gene interactions in erythropoiesis [15] with the inferred GRN, demonstrated that out of 16 known interactions, 11 were captured by IQCELL (over 68%), (**Fig 6D**). Among the 5 uncaptured interactions, 3 of them are related to two genes, *Fli1* and *Spi1* which are known to be activated in the megakaryocyte lineage [15], and thus their effects may be masked in erythropoiesis. To initiate the simulated dynamics of the GRN, we programmed the initial state to resemble the binarized gene expression of MEPs (**S6A Fig**). Simulation results show that the dynamics matches the progression of scRNA-seq data (**Figs 6E** and **S6C and S6D**) and the corresponding attractor state matches the ErP stage (**S6E Fig**).

Next, we tested the effect of gene perturbations on simulated developmental trajectories (**Fig 6F**). The TF *Gata1* plays an important role in erythropoiesis as it regulates multiple downstream genes in erythroid cell development [15]. *In vitro* and *in vivo* studies of hematopoietic cells that lack the normal expression level of *Gata1*, show that the lack of *Gata1* halts erythroid differentiation [48,49]. Our model also predicts a block of erythroid differentiation in the

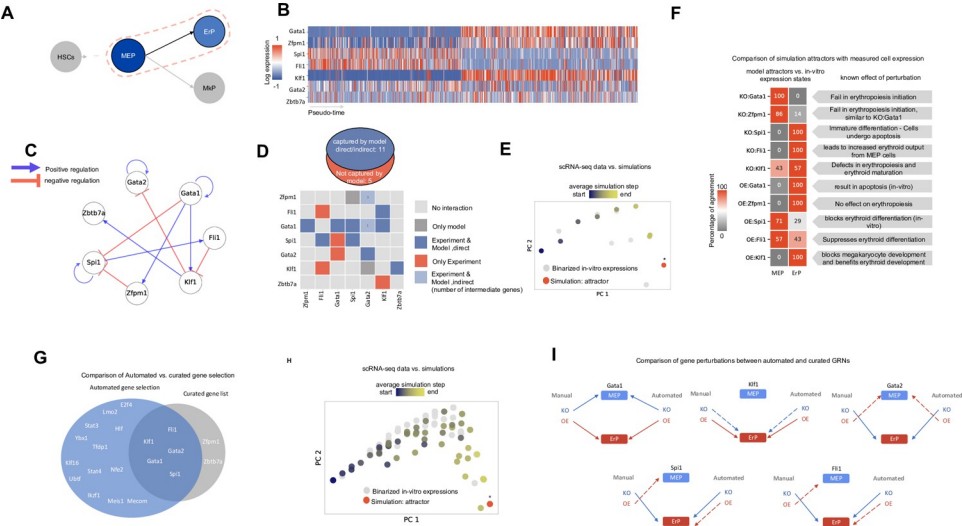

**Fig 6. Constructing mouse erythropoiesis GRN.** (A) Summary of the scope of the sc-RNA seq data used as an input to IQCELL [17]. Erythroid progenitors (ErP) arise from megakaryocyte/erythroid progenitors (MEPs). (B) Log transformed expression matrix for selected genes from sc-RNA data along the pseudo-time axis. (C) The constructed GRN for mouse erythropoiesis. The GRN is obtained by constraining the possible interactions to both follow the *in vitro* data progression when executed as a logical network and maximize mutual information between gene pairs. Positive and negative interactions are represented by blue and red edges, respectively. (D) Out of 16 known interactions [15], 11 were captured by IQCELL (top). Detailed representation of the proposed interactions by IQCELL and experimentally reported ones. Rows and columns represent regulators and effector genes, respectively. Blue indicates that the interaction is captured by the model directly (dark blue) or indirectly (light blue); in the latter case, the numbers indicate the number of intermediate genes. Dark gray indicates that the interaction is only proposed by IQCELL. The red color indicates the experimentally validated interaction is not present in the model. Light gray cells indicate no interaction (bottom). (E) The PCA plot of the simulated developmental trajectories are overlaid on the binarized scRNA-seq (the curated genes). The detected attractor is colored red and is marked by star (*). The arrow (blue-red) represent the direction of inferred pseudo-time. (F) Percentage of similarity between the model attractors under perturbations (vertical axis) and binarized expression of MEPs and ErPs (horizontal axis) (left). Description of known effect of the gene perturbation (right). (G) Comparison of automated vs curated TF selection show that the TF selection pipeline captures 5 of 7 genes in the curated gene list. (H) The PCA plot of the simulated developmental trajectories are overlaid on the binarized scRNA-seq (the genes resulting from automated TF selection). The detected attractor is colored red and is marked by star (*). The arrow (blue-red) represent the direction of inferred pseudo-time. (I) Comparison of gene perturbations between automated and curated GRNs show 8 matches of the predicted states out of 10 perturbations.

absence of *Gata1* with a single attractor state resembling the MEPs (**Fig 6F**). Notably, while IQCELL predicts that OE of *Gata1* leads to the ErP cell state, *in vitro* assays show that OE of *Gata1* causes apoptosis [50]. *Zfpm1* (*FOG-1*) is a *Gata1* transcriptional cofactor, it has been shown that the lack of *Zfpm1* leads to arrested development in erythropoiesis at a stage similar to *Gata1* KO cells [51]. Our model also predicts that *Zfpm1* KO cells fail at an early stage (**Figs 6F** and **S6F and S6G**). The OE of *Zfpm1* looks to have no significant effect in erythropoiesis [52] which is consistent with our *in silico* result (**Fig 6F**). The deficiency of *Spi1* in erythroid progenitors leads to cell differentiation which is consistent with IQCELL KO simulation result (**Fig 6F**), but also it has been reported that because of immature differentiation, the cells undergo proliferation arrest [53]. On the other hand, the IQCELL model predicts that OE of Spi1 stops erythroid differentiation at an early stage (**Figs 6F** and **S6G**), it has been reported that the OE of *Spi1* leads to blockage of erythroid differentiation *in vitro* [54]. We have also simulated the KO effect of *Fli*, which leads to increased erythroid output [55], the OE effect of *Fli*, which suppresses the erythroid differentiation [56], KO effect of *Klf1*, which causes defects in erythroid differentiation [46], and the OE effect of *Klf1*, which blocks megakaryocyte

differentiation and increases the erythroid differentiation output [57]. These *in silico* results show consistency with experimental evidence (**Figs 6F**, more details at **S6F**).

### Testing IQCELL performance via automated TF selection in erythropoiesis

We also tested IQCELL when candidate GRN TFs were selected algorithmically from scRNA-seq data (**S7A Fig**). Given the GDS, we selected the top 18 TFs to be included in the model (**S7B Fig**). The gene list produced by the TF selection algorithm captured 5 out of 7 TFs that were included in the curated model (*Gata1*, *Gata2*, *Spi1*, *Fli1*, and *Klf1*), (**Fig 6G**). Next, we applied IQCELL's GRN inference pipeline to construct a functional GRN for the system (**S7C–S7F Fig**). The result of the dynamic simulation of the GRN matched the observed dynamics in scRNA-seq data (**Figs 6H** and **S7G and S7H**) with the attractor state matching the ErP state (**S7I Fig**).

After performing systematic KO and OE simulations on the GRN (**S7J and S7K Fig**), we compared IQCELL's predictions on gene perturbations between the GRN that is made via curated gene list and automated TF selection (**Fig 6I**). Comparing the predictions of two GRNs shows that out of 10 common perturbations, the results agreed between 8 of them. The only differences are the outcome of OE of *Spi1* and *Fli1* (**Fig 6I**).

Finally, we compared IQCELL's performance with other state of the art GRN inference methods [12] using both the T-cell and erythropoiesis datasets. Our results show that IQCELL outperformed the other methods, with a higher area under the precision-recall curve (AUPRC) for both datasets (**S9 Fig**).

## Discussion

There is an increasing availability of scRNA-seq datasets for different developmental systems. To date, the inference, analysis, and simulation of logical GRNs directly from scRNA-seq data have not been integrated together. Here we present IQCELL, an integrated platform implemented as a Python package to infer, analyze, and simulate GRNs directly from scRNA-seq data and pseudo-time order of the cells.

IQCELL is able to capture, directly from scRNA-seq data, over 74% (38 out of 54) of the reported gene interactions in early T-cell development and erythropoiesis [15,16]. These interactions were obtained and characterized by decades of research and experiments [30]. For example, regulators of *Bcl11b*, an essential gene for T lineage commitment, were successfully identified by IQCELL. More than half of the interactions that were not captured in T-cell development by IQCELL were *Spi1* effector genes, which is a Notch signaling antagonist [58]. However, Notch signaling contains *Spi1* inhibitory effect on T-cell regulators [58] and potentially masks *Spi1* negative regulatory roles in early T-cell development.

We also tested the dynamics of the obtained GRNs. For example, in early T-cell development, we showed that when the logical GRN is simulated from the ETP cell state, its dynamics evolves to the cell state associated with the DN3 stage, in agreement with experimental observations. Importantly, we showed that our platform can produce GRN models with high predictive power for the effect of genetic perturbations. For example, simulated KO of *Bcl11b* caused the developmental trajectory to halt at the DN2 stage, in agreement with experimental studies in early T-cell development. We identified eight gene perturbations that halt T-cell development at different points between the ETP and DN3 stages, and IQCELL showed satisfactory agreement for all perturbations with experimental studies (**Fig 4C**). We have also tested IQCELL against gene perturbations with known effects in erythropoiesis with satisfactory agreement (**Fig 6F**).

In many biological systems, curated and validated lists of important genes are not *a priori* known. IQCELL's customized pipeline for TFs selection was used to independently create a new set of TFs for mouse T-cell development and erythropoiesis. We show that this inference module successfully captured many of the known important TFs in both systems (**Figs 5B** and **6G**). We used this set of genes to create new GRNs and test the robustness of IQCELL's prediction against gene perturbations. For example, in early T-cell development, we showed that out of 18 perturbations that were in common between the two GRNs (automated and curated), IQCELL predictions were similar in 17 cases. This validation encourages future investigations into systematic gene perturbation outcomes of this GRN.

These results show that the multi-step strategy implemented in IQCELL is effective for reconstructing functional GRNs from existing information in scRNA-seq data. Because its methodology is not specific to a single developmental system, IQCELL may be broadly useful in understanding how GRNs contribute to cell development in a variety of developmental contexts. IQCELL results may help uncover functional relations between genes and thereby help design more effective gene manipulation strategies to drive cell differentiation cultures *in vitro* toward fates of interest. Synthetic gene interactions can be added to the GRNs outputted by IQCELL to predict the effect of novel synthetic gene circuits on native cell GRNs. A major goal in systems biology is the creation of multi-scale models that connect the decisions of individual cells within a multicellular system to emergent properties of the whole tissue [59,60]. IQCELL can fill an important layer in such multi-scale models. By exposing the intracellular decision-making machinery of single cells, IQCELL could interface with other methods that connects these cellular decisions to tissue-level dynamics.

To date, there have been many methods introduced for reconstructing GRNs from single-cell data [12] and many of them focus on finding some type of correlation between genes. In one case, binarized sc qPCR data was used to decode logical GRNs for embryonic blood development [14]. In another recent study, sc qPCR data and its pseudo-time order was used to decode GRNs of blood stem cells [13]. However, to the best of our knowledge, prior to IQCELL there has not been any existing method or platform to infer executable and logical GRNs from scRNA-seq data, nor have previous methods dealt with the associated challenges of lower sensitivity and dropout effects.

Although the IQCELL framework allowed us to effectively model regulatory modules as logical (Boolean) gates where no extra parameters are required, logical models are limited in some respects. Firstly, logical modeling cannot effectively capture dose-dependency in gene interactions; for example, it is known that the downstream responses to *Gata3* are dose-dependent [61]. We suggest in future that this aspect be captured by multilevel models [62]. Multi-level modeling requires more sensitive measurements of TF expression levels, which may become feasible with emerging TF profiling methods [63]. Secondly, capturing biophysical timescales in the logical framework is not trivial; one solution would be assigning a weighted time scale [64] to each simulation update step of the logical model. This can potentially help to include some time-scale sensitive events in cell GRN dynamics, such as stochastic chromatin restructuring events [37].

Here we have tested IQCELL's performance in single developmental trajectories. However, IQCELL could be used to infer GRNs from multiple developmental trajectories (e.g. trajectories with bifurcations) by inferring candidate GRNs for each trajectory. Since IQCELL provides users with a flexible framework, future studies could integrate other sources of information such as binding of TFs to DNA via ChIP-seq [65] and CRISPR screening on the effect of gene perturbations on developmental trajectories [66] to potentially improve GRN reconstruction. In a recent study, the combination of scATAC-seq and scRNA-seq with machine learning methods have been used to infer a set of informative TFs during

differentiation [67]. In addition, new opportunities are arising to investigate the decision making machinery of the cells in their native environment (via in situ cell profiling) [68]. The combination of these methods, prior knowledge of cell-cell interactions [69,70], and emerging theoretical knowledge and computational technologies for capturing and quantifying spatio-temporal information content of cell signaling [71–74] can be used as invaluable resources for the next generation of GRN inference methods. These next-generation methods would ideally integrate cell signaling [75] with GRNs directly from multi-omics single-cell data. In conclusion, the results presented here suggest that IQCELL will be a broadly useful tool to study cellular decision making in a variety of developmental systems.

## Methods

### Selecting a suitable scRNA-seq dataset as IQCELL's input

There is a direct relationship between the quality of the scRNA-seq data (and the inferred pseudo-time) with the quality of the inferred GRN. A suitable dataset for IQCELL captures the cells in transient states (i.e., along developmental trajectories or other interesting biological transitions) so IQCELL can correctly infer the dynamics of the fate changes. For example, some *in vitro* differentiation processes can take several days to happen, and having samples from intermediate days will potentially capture the transient cells. Also, more depth in sequencing will give more confidence in counting TFs, and potentially better performance of IQCELL as the accuracy of calculating mutual information and update rules will increase.

### Gene expression recovery

scRNA-seq data is usually affected by dropouts, which is a technical term that is used to describe the false-negative reads of mRNAs. Dropout effects cause the expression profile of genes to be underrepresented. Usually, genes with low copy numbers (e.g TFs) are more impacted by this effect. IQCELL applies a recent method (MAGIC) that uses a graph-based imputation method to infer the expression of dropouts [24] (**S1B Fig**). The MAGIC algorithm was selected based on a recent benchmarking paper [76] as it outperforms other methods. This imputation is important as dropouts can affect the inference of gene relations (**S1B Fig**). In summary, MAGIC calculates the affinities between neighboring cells in high dimensional data and uses this information to recover the undercounted values of gene expression of individual genes (dropouts) from the neighboring cells. This process leads to the imputation of gene expression [24].

### Selecting a curated gene list for early T-cell differentiation

Selecting a small subset of genes to include in a functional GRN from the entire set of genes detected by scRNA-seq is, in general, a challenging task. Fortunately in mouse T-cell development, many relevant genes are known. We have curated a list of genes based on literature [16,31,37], and limited ourselves to genes that are included in to HVGs. Below, we describe a possible general approach for a selecting smaller subset of genes from a large set.

Selecting relevant genes for a functional/causal GRNs is generally a multilayered process (**S1A Fig**) which ideally combines multiple sources of information. One possible source is prior knowledge of important genes in the process, which can be obtained through literature review or systematic gene perturbation experiments (such as genome-wide CRISPR screening). Alternatively, genes of interest can be selected directly from scRNA-seq data via various information theory metrics [77]. Many scRNA-seq data analysis packages set an initial filter to only select HVGs for downstream analysis. HVGs are genes whose mean-scaled variance

exceeds an automatic threshold. Although this is generally a useful filter, it does not necessarily select for genes whose expression levels vary significantly across pseudo-time. Therefore, IQCELL has a built-in function to visualize expression dynamics along pseudo-time and calculate the degree of variation, which can be used as an additional input for gene selection.

Besides these, there are other network-based approaches to select informative genes. These methods typically prioritize genes with connections to many other genes (high degree). Finally, enrichment analysis and ChIP-seq data can be another source of gene selection; however, these methods are generally low-throughput, noisy, and prone to false positives and negatives. For this study, we manually curated a list of genes based on biological significance [16] and dynamics along the pseudo-time (**S1A Fig** and **S1 Table**).

## Creating an initial list of TFs in automated gene selection

The IQCELL gene selection pipeline starts with a set of HVGs. Next, it uses the pySCENIC platform [23], which was chosen as it is the only platform to our knowledge that effectively combines binding motifs and expression enrichment [78]. We select TFs that are components of core regulons during the differentiation process to calculate their GDS.

## Gene dynamics score

The GDS measures the variability of genes along the pseudo-time. The method implemented in IQCELL for this purpose is based on the mean value. The normalized gene expression along the pseudo-time is assigned to a number of bins (5 as default).

For each gene (g), the mean of each bin is calculated ($MB_i^g$), and the matrix of mean distances (MMD) are formed:

$$\mathrm{MMD}_{i,j}^g = |\mathrm{MB}_i^g|$$

The GDS for each gene is calculated as the maximum of arguments of MMD matrix:

$$\mathrm{GDS}^g = \mathrm{argmax}(\mathrm{MMD}_{i,j}^g)$$

## Establishing the initial gene-gene interaction network

To form the initial gene-gene interaction network from scRNA-seq data (**S1C Fig**), IQCELL first forms a list of all possible pairwise gene-gene interactions. This list does not include auto-regulation by default (optional) except in some cases (see **Implementing the Z3 reasoning engine to infer logical GRNs**). Next, it uses a recent method (DREMI) to calculate the resampled and conditional mutual information between gene pairs [26] (**S1C Fig**). In general, the mutual information (I) between a pair of variables X and Y (where X and Y represent two genes) is calculated as below:

$$I(X; Y) = H(Y) - H(Y|X)$$

where H(X) is called the entropy of distribution X and P(X) is the probability density of the variable X:

$$H(X) = \sum P(X)\log P(X)$$

The mutual information score is symmetric and unsigned. Next, IQCELL applies the Pearson correlation coefficient to assign a sign (+/- which represents activation/repression) to the interactions, based on the sign of the correlation between two genes (**S1C Fig**):

$$c(X, Y) \geq 0 \rightarrow +$$

$$c(X, Y) < 0 \rightarrow -$$

Where c(X,Y) is the Pearson correlation coefficient of the pair of gene X and Y:

$$c(X, Y) = \frac{E(XY) - E(X)E(Y)}{\sigma_X \sigma_Y}$$

E(X) is the expected value and $\sigma_X$ is the standard deviation of X. Finally, IQCELL eliminates interactions that are difficult to assign a sign (+/-) to them (their absolute correlation values are smaller than a threshold). The default threshold is permissive so that many interactions can pass and will be signed (for example see **S1C Fig**). If an update rule is found for a particular gene, lowering the threshold will only increase the search-time, because it is likely that the link will be selected (due to higher correlation). This threshold allows us to assign a sign (positive or negative) to interactions, its initial purpose is not to cut interactions, but to eliminate interactions with sign ambiguity.

## Binarization of gene expressions

IQCELL binarizes the expression (**S4F Fig**) for each gene individually. To do this, it can apply two possible methods that can be chosen by the user (**S1D Fig**). The first method finds the mean of expression and assigns 'off' (0) to the genes (for each cells) with the expressed numbers of mRNAs that are smaller than the prefixed threshold and 'on' (1) if they are larger. The second method (the default method) binarizes the expression based on the cut-off identified by k-means clustering (k = 2) method (**S1D Fig**) [27]. In general, we group the expression of each gene in each cell $g_i$ (i∈$N_{cells}$) into two possible groups $S_i$ = {$S_{off}$, $S_{on}$} by finding:

$$\arg \min_S \sum_{i \in off, on} \sum_{g \in S_i} ||g - \mu_i||$$

In which $\mu_i$ (centroids) is the mean of expression values in 'on' or 'off' cluster. The threshold ($\tau$) is obtained as the average of two centroids:

$$\tau = (\mu_{off} + \mu_{on})/2$$

The binarized gene expression value ($G_i$) is obtained similar to the mean method:

$$if(g_i \leq \tau) \rightarrow G_i = 0$$

$$if(g_i > \tau) \rightarrow G_i = 1$$

## Establishing the gene hierarchy via pseudo-time

After binarization of expression levels, IQCELL calculates an interaction hierarchy to further filter gene-gene interactions, thereby eliminating interactions that are less likely to be causal. First, it averages the binarized expression values in a sliding window over pseudo-time (**S1E Fig**). To do this, IQCELL first sorts the cells based on their pseudo-time values $c_i$ (i∈$N_{cells}$). Next, it averages the values of binarized gene expressions over an averaging window (with the default length of L = $N_{cells}/N_{genes}$). This results in a density representation of the binarized gene expression values along pseudo-time (t) (**S1E Fig**):

$$D(t) = \sum_{i \in (t-L/2, t+L/2)} G_i/L$$

Next, it calculates transition points between low-to-high or high-to-low density regions for all genes (**S1E Fig**) and sorts genes based on their transition points (**S1E Fig**). Finally, IQCELL includes autoregulation (autoactivation) as a possible self-interaction for genes with less than two possible activators. This step is optional and is used in the current study, leading to inclusion of autoactivation interactions for *Hes1*, *Notch1*, *Lmo2*, and *Spi1* in the final interaction network.

## Implementing the Z3 reasoning engine to infer logical GRNs

To generate functional GRNs, IQCELL implements a modified network inference strategy in the Z3 engine [13]. This method effectively finds optional logical rules for each gene based on the possible list of interactions obtained from previous steps. The optimal rules are those that when executed as logical gates for each gene (given the state updates of other genes along the pseudo-time as an input), follow the experimental data. This is quantified with the percentage of similarity (based on Hamming distance) between the two (**S1F Fig**). Similar to [13], we allow up to four possible activators and up to two repressors for each gene. In contrast to [13] and similar to [8], for gene activation, we assume that all the activators are necessary (which is implemented with the 'and' logic gate), but only one repressor is enough for repression (which is implemented with the 'or' logic gate). In summary, the most general logical rule for the regulation of a gene ($g_j$) by (the maximum number of) six regulators including (maximum) four activators ($A_1$, $A_2$, $A_3$, $A_4$) and (maximum) two repressors ($R_1$, $R_2$) is:

$$g_j = (A_1 \text{ and } A_2 \text{ and } A_3 \text{ and } A_4) \text{ and not } (R_1 \text{ or } R_2)$$

Which indicates that all activators and none of the repressor should be expressed for the gene to be expressed. The assumption of having 'and' between activators and 'or' between repressors is a strict assumption. IQCELL has the option to relax this assumption to include both 'and' and 'or' in all rules with the cost of more computational time. We suggest that users first use the default configurations, and in the case of not finding any satisfying rule, then try the relaxed assumptions.

The maximum number of 6 regulators in total is selected due to the computational cost of searching the state-space of possible interactions (**S8A Fig**). In summary, the process of finding the rules is costly, since for each possible rule, we calculate the agreement score for thousands of cells along the pseudo-time. Having up to 4 activators and up to 2 repressors, gives rise to up to nRules, where this is defined by (**S8A Fig**):

$$nRules = \left[ \binom{N}{1} + \binom{N}{2} + \binom{N}{3} + \binom{N}{4} \right] \times \left[ \binom{M}{0} + \binom{M}{1} + \binom{M}{2} \right]$$

Where N and M are the numbers of all possible activators/repressors in the initial interaction network respectively. Increasing the possible value of activators/repressors will increase $nRules_T$ and computational cost. As a reference, we have included the distribution of edge degree for all genes of the final Boolean GRN in this study (**S8B Fig**).

Next, for each gene, the rule with the highest average mutual information for interactions is selected by IQCELL for the final GRN (**S1 Table**). Inclusion of self-activation in the final Boolean GRN model selection will be costly for the update rule since we have set the mutual information of self-activation interaction to '0'. This reduces the average mutual information weight of terms that have self-activation. This decision is made because self-activation cannot be weighted effectively by mutual information value against other genes (mutual information is the final criteria used in IQCELL to select the final GRN). However, including self-activation in a Boolean update rule can help the rule to make the cutoff of Z3 step. This ensures that the self-activation is only included if necessary.

## Asynchronous simulator of GRN under normal and perturbed condition

To analyze the system level behavior of the obtained GRN models and predict the effect of gene perturbations on developmental trajectories, IQCELL uses asynchronous Boolean simulations. Boolean GRNs contain a set of genes $\mathbf{G} = \{G_1, G_2, \dots G_{N_{genes}}\}$ and their update function which encodes the gene regulatory details (one update function per gene). The update function of a gene ($F_{G_i}$) implies what will be the activity state of that gene (on or off) at the next discrete time point $G_i(t+1)$ given the state of all the genes at the current discrete time point $\{G_1(t), G_2(t), \dots G_{N_{genes}}(t)\}$:

$$G_i(t+1) = F_{G_i}(G_1(t), G_2(t), \dots G_{N_{genes}}(t))$$

IQCELL uses the asynchronous update strategy [7]. In this strategy, at each discrete time step, only one random gene is selected and updated. This results in stochastic dynamics. This lets us average the expression states (average_exp) at each discrete time point (j) over the ensemble of stochastic states that started from the same initial point and are now at that particular time point ($\mathbf{EX}$(j)) (**Fig 3G**):

$$\text{average\_exp}_j = \sum_{S \in EX(j)} S/\|EX(j)\|$$

Where S = {0,1} therefore:

$$0 \leq \text{avrage\_exp}_j \leq 1$$

To calculate the average step of each simulation data point (e.g. **Fig 3F**), we first find the attractor of interest (the attractor marked by $*$), then for each simulation data point (p), we find where that particular point is ($X_T^P$) for each possible trajectory, then we normalize this value to the length of the trajectory ($L_T$):

$$x_T^P = X_T^P/L_T$$

Where $x_T^P$ is the scaled place of the point P in trajectory T. Finally for each point, we average over all trajectories:

$$asp^P = <x_T^P>_T$$

which gives us the average simulation step ($asp^P$) for each point.

IQCELL also has a built-in function to perturb the GRNs in two ways. It has the capability to perform KO (setting the gene to be always 'off') and OE (setting the gene to be always 'on') experiments simultaneously for one or multiple genes systematically. Also it can perturb the gene-gene interactions systematically (**S2 Fig**).

## Software architecture of IQCELL

IQCELL is implemented as a Python package. It is modular and scalable, to help researchers to expand, optimize, and customize it for their future studies. Also, it has a minimal Python interface, which allows the users with minimal computational skills to use it, and implement it in their system of interest.

## Pre-processing the scRNA-seq data

To dissect the developmental trajectory of T-cell development (from ETP to DN3 stages), we used an existing scRNA-seq dataset [18], and analysis pipeline [79]. After the quality control

and normalizing expression values, single-cell transcriptional states were visualized in reduced dimensional space using UMAP (**S4A Fig**). To understand the underlying structure of data we performed clustering based on the Louvain method (**S4B Fig**) which yields 14 sub-clusters. Next, we evaluated the expression pattern of developmentally important genes in blood and particularly T-cell development. ETP associated genes (*Flt3*, *Lmo2*, *Mef2c*) are all expressed in the cell clusters in the left side (**S4C Fig**). DN2A stage is marked by *Il2ra*, this gene along with the gene associated with committed DN2 cells (*Bcl11b*) and DN3 associated genes (*Cd3e* and *Rag1*) expression are localized on the right side (**S4C Fig**). Altogether we conclude that cluster 0 includes many of the cells at the ETP stage and the developmental progression is toward clusters 9 and 11 as the endpoints (**S4C Fig**).

Using a previously reported approach [18], we performed pseudo-time ordering on the subset of genes that are known to be developmentally important in T-cell development or are alternate lineage markers. The selected genes were similar to [18] and DDRtree pseudo-time ordering was performed on the data [22]. As established in the cluster analysis, step cluster number 0 is the best candidate as the cluster with the earliest developmental stage and is used as the root for the algorithm. The result shows a single trajectory starting at cluster 0 and progressing toward later stages (**S4D Fig**). Pseudo-time ordering shows the dynamics of genes as a function of differentiation (**S4E Fig**).

We processed the erythroid differentiation dataset the same way as in the original publication [17] to better support our comparisons; specifically we used the Slingshot method to infer the developmental trajectory [80].

## Calculating area under the precision-recall curve (AUPRC) for IQCELL outputs

To calculate the area under the AUPRC, we followed Pratapa et al., 2020. For each dataset, we first weighted the gene interactions based on the average mutual information of the rule that the interaction is located in. For duplicated interactions that are present in two or more rules, we picked the highest rank. Next, we weighted the interactions based on their position in the raking and used the Beeline software [12] to calculate AUPRC for IQCELL and other the methods (**S9 Fig**).

## Sample preparation and single-cell RNA-sequencing of *in vitro* T-cell differentiation

Isolated fetal liver cells from decapitated E13.5 CD1 mouse embryos were subjected to TER-119 depletion by EasySep magnetic sorting (STEMCELL Technologies). Next, sorted HSPCs (Sca-1+ cKit+) cultured at 3.1 x10E3 HSPCs/cm2 (corresponding to 1000 cells/well) in DL4 (10 μg/mL) and VCAM-1 (2.32 μg/mL) coated 96-well plates [81]. 10X Chromium was used to prepare single-cell cDNA libraries, and Illumina Nextseq was used to 3' sequence the samples. Gene-barcode expression matrices were calculated from the raw data via CellRanger (10X Genomics).

## Supporting information

**S1 Fig. Overview of IQCELL parts and algorithms.** (A) Overview of gene selection criteria (left). IQCELL's gene selection pipeline (right). After selecting HVGs, IQCELL uses pySCE-NIC to select active regulons (TFs and their effectors), and finally IQCELL uses GDS to rank and select TFs for the final list. (B) Overview of gene expression recovery step. The scRNA-seq data is corrected for dropout effect via a borrowed library 'MAGIC' from literature (left). Raw

(grey) and recovered (blue) expression of Bcl11b vs Gata3 (right). (C) Overview of generating the initial interaction network. The steps toward obtaining the directional and signed interaction network from an initial list of genes (left column). The histogram of mutual information between gene pairs, the histogram of Pearson correlation between gene pairs (the correlation value of 1 is for the correlation of genes with themselves, marked by red), and mutual information vs. correlation values (right column). In the histogram of Pearson correlation values, the interactions that end up in the final model are marked by short red lines and the vertical green line shows the sign cutoff (which its value has been chosen to be permissive); results are for the T-cell dataset. (D) Overview of expression binarization step. There are two implemented binarization methods in IQCELL. K-means clustering (default) and binarization based on the mean value of expression of the gene between all the cells (left). Example of binarization of genes and their expression along the pseudo time (right). (E) Overview of generating the gene hierarchy step from the binarized gene expressions. First, the expression levels are averaged with a sliding window along the pseudo-time. This results in the density profile of binarized genes along the pseudo-time (left). Next, based on clustering the density, the transition point (from high to low or low to high) are captured (center). Finally, genes are sorted based on transition points. Genes can interact with genes with a lower rank (right). (F) Overview of implementing the Z3 reasoning engine. At this step, the filtered set of interactions are used to make provisional update rules for each gene.
(PDF)

**S2 Fig. GRN perturbations via IQCELL.** (A) Systematic gene perturbations. The expression states of the model attractor (left). The percentage of similarity between the attractors (vertical axis) and the binarized expressions of CLP, ETP, DN2A, DN2B, and DN3A cells (horizontal axis) [38] (right) for systematically perturbed GRN with single gene KO and OE. (B) Systematic gene-gene interaction perturbations. Overview of GRN link perturbation (top). The percentage of similarity between the attractors (vertical axis) and the binarized expressions of CLP, ETP, DN2A, DN2B, and DN3A cells (horizontal axis) [38] (bottom) for systematically perturbed GRNs.
(PDF)

**S3 Fig. Implementation of IQCELL with another early T-cell development scRNA-seq data set.** (A) To demonstrate the universality of IQCELL, we have tested this platform with another in-house scRNA-seq data. The IQCELL tutorial in the IQCELL website is based on this dataset. We used 10X scRNAseq by performing whole-genome transcriptional analysis. These experiments are performed with the mouse T-cell progenitor populations from fetal liver (FL) hematopoietic stem and progenitor cells (HSPCs) differentiated *in vitro* using the DL4 +VCAM platform [81]. FL HSPCs were seeded on DL4+VCAM coated plates and cultured for 4 or 7 days prior to analysis, or immediately sorted and captured for library preparation. Pooling cells from multiple differentiation time points enabled sampling of cells from the entire T-cell lineage progression, rather than just endpoint transcriptional states. Here we have selected a small set of 6 genes that are important in the early T-cell development (from ETP to DN2 stages). The heat map shows the expression matrix of smoothed binarized expressions along the pseudo-time. (B) The set of all possible gene-gene interactions, filtered by interaction hierarchy and mutual information cut-off (the thickness of lines represents the mutual information between genes), and signed by correlation. (C) Provisional GRN for early mouse T-cell development. (D) Detailed representation of the proposed interactions provided by IQCELL and experimentally reported ones. (E) PCA plot of the binarized scRNA-seq data color coded with the pseudo-time values attributed to each cell. The binarization is performed by clustering the scRNA-seq expressions into expressed or not expressed levels. On top of that, the binarized

expressions of CLP, ETP, DN2A, DN2B, cells have been calculated from the Immgen microarray data [38] and overlaid on RNA-seq data. (F) PCA plot of the simulated developmental trajectories are overlaid on the binarized scRNA-seq. (G) Expression states of the model attractors (top). The percentage of similarity between attractor and known cell states [38]. (H) Averaged gene expression of the simulated data at each simulation step. All simulations started from the same initial condition.
(PDF)

**S4 Fig. The overall workflow of preprocessing the scRNA-seq data. (A)** UMAP representation of data overlaid with sample ID, number of genes per cell, and density plot. (B) Clustering of scRNA-seq data. (C) Gene expression of stage-specific gene overlaid on top of UMAP. (D) The pseudo-time trajectory of the data inferred by Monocle platform. (E) Example expression of genes along the pseudo-time trajectory. (F) The normalized and scaled to 1 expression of some example genes.
(PDF)

**S5 Fig. The GRN created based on the gene selection pipeline for T-cell development. (A)** The GDS of genes, the top 14 genes have been selected (left). Log transformed expression matrix for selected genes from sc-RNA data along the pseudo-time axis. Gene expression is corrected for dropout effects using MAGIC. Red indicates high expression, blue indicates low expression (right). **(B)** Smoothed binarized gene expression matrix (expression density). Gene expression values were binarized by clustering, averaged over a pseudo-time window, then sorted based on transition points from early to late. Red indicates high expression, blue indicates low expression. **(C)** The set of all possible gene-gene interactions, filtered by interaction hierarchy and mutual information. **(D)** The provisional GRN for early mouse T-cell development. **(E)** Average gene expression at each simulation step. All simulations started from the same initial condition (ETP) and move toward the same attractor (*). **(F)** Systematic KO: The percentage of similarity between the attractors (vertical axis) and the binarized expressions of CLP, ETP, DN2A, DN2B, and DN3A cells (horizontal axis). **(G)** Systematic OE: The percentage of similarity between the attractors (vertical axis) and the binarized expressions of CLP, ETP, DN2A, DN2B, and DN3A cells (horizontal axis)
(PDF)

**S6 Fig. GRN created based on the red blood cell development dataset.** (A) Average gene expression at each simulation step. All simulations started from the same initial condition (MEPs) and move toward the same attractor (*). (B) The set of all possible gene-gene interactions, filtered by interaction hierarchy and mutual information (and signed by correlation. Positive and negative interactions are represented by blue and red edges, respectively. Edge width represents the relative amount of mutual information of the interaction. Nodes colored red have higher out-degrees. (C) The PCA plot of the binarized scRNA-seq data color coded with the pseudo-time values attributed to each cell. The binarization is performed by clustering the scRNA-seq expressions into expressed or not expressed levels. (D) Average gene expression at each simulation step. All simulations started from the same initial condition (MEPs) and move toward the same attractor (*). (E) Expression states of the GRN model steady state attractors. Genes that are expressed (1) and not expressed (0) are represented with blue and grey squares, respectively (left). Percentage of similarity between the model attractors under perturbations (vertical axis) and binarized expression of MEPs and ErPs (horizontal axis) (right). (F) Systematic gene perturbations. The expression states of the model attractor (left). The percentage of similarity between the attractors (vertical axis) and the binarized expressions. (G) The PCA plot of the simulated (perturbed) developmental trajectories are overlaid

on the binarized scRNA-seq. The simulated data is color coded by the value of average simulation step (average distance to the attractor of simulation).
(PDF)

**S7 Fig. The GRN created based on the gene selection pipeline for red blood cell development.** (A) Overview of gene selection procedure for erythropoiesis data. (B) The GDS of genes, the top 14 genes have been selected. (C) Log transformed expression matrix for selected genes from sc-RNA data along the pseudo-time axis. Gene expression is corrected for dropout effects using MAGIC. Red indicates high expression, blue indicates low expression. (D) Smoothed binarized gene expression matrix (expression density). Gene expression values were binarized by clustering, averaged over a pseudo-time window, then sorted based on transition points from early to late. Red indicates high expression, blue indicates low expression. (E) The set of all possible gene-gene interactions, filtered by interaction hierarchy and mutual information. (F) The provisional Boolean GRN for early mouse T-cell development. (G) The PCA plot of the binarized scRNA-seq data color-coded with the pseudo-time values attributed to each cell. The binarization is performed by clustering the scRNA-seq expressions into expressed or not expressed levels. (H) Average gene expression at each simulation step. All simulations started from the same initial condition (MEPs) and move toward the same attractor (*). (I) Expression states of the GRN model steady state attractors. Genes that are expressed (1) and not expressed (0) are represented with blue and grey squares, respectively (top). Percentage of similarity between the model attractors under perturbations (vertical axis) and binarized expression of MEPs and ErPs (horizontal axis) (bottom). (J) Systematic KO: The percentage of similarity between the attractors (vertical axis) and the binarized expressions of MEPs and ErPs (horizontal axis). (K) Systematic OE: The percentage of similarity between the attractors (vertical axis) and the binarized expressions of MEPs and ErPs (horizontal axis).
(PDF)

**S8 Fig. Possible update rules for each gene.** (A) Calculating the number of possible rules for the update function of a gene. (B) Histogram of number of activator/repressors of all genes.
(PDF)

**S9 Fig. Comparing IQCELL performance with other methods.** Comparison of the area under the precision-recall curve of IQCELL with other standard GRN inference methods (Pratapa et al., 2020). IQCELL shows improved performance over other methods for both the T-cell and the erythropoiesis datasets.
(TIF)

**S1 Table. The final gene list and provisional logical GRN for early T-cell development.** The rules are picked from possible rules in Z3 step based on maximizing the average mutual information per gene.
(PDF)

**S2 Table. Genes used for supervised trajectory inference.**
(PDF)

**S3 Table. Comparison of attractors of perturbed GRN with the known cell states from microarray data.**
(CSV)

**S4 Table. The attractor states of perturbed GRN.**
(CSV)

**S5 Table. The final gene list and provisional logical GRN for erythropoiesis.**
(PDF)

## Acknowledgments

We thank Sara-Jane Dunn, Boyan Yordanov, and Ellen V. Rothenberg for our fruitful discussions and Yale S. Michaels and John M. Edgar for critically reading the manuscript. We also thank Microsoft Research (Cambridge, UK) and Sara-Jane Dunn for facilitating the opportunity for the author to deepen his understanding of the Z3 reasoning engine. We thank Ellen V. Rothenberg, and Bertie Gottgens for generating publicly available high-quality scRNA-seq data sets that have been used in this study.

## Author Contributions

**Conceptualization:** Tiam Heydari, Peter W. Zandstra.

**Investigation:** Matthew A. Langley, Cynthia L. Fisher, Shreya Shukla, Michael Hughes.

**Methodology:** Tiam Heydari.

**Resources:** Kelly M. McNagny.

**Software:** Tiam Heydari, Matthew A. Langley, Ayako Yachie-Kinoshita.

**Supervision:** Peter W. Zandstra.

**Writing – original draft:** Tiam Heydari, Daniel Aguilar-Hidalgo, Peter W. Zandstra.

**Writing – review & editing:** Tiam Heydari, Matthew A. Langley, Cynthia L. Fisher, Daniel Aguilar-Hidalgo, Kelly M. McNagny, Peter W. Zandstra.

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
