## [Decision Letter · Decision Letter 0]

28 Sep 2021

Dear Prof. Zandstra,

Thank you very much for submitting your manuscript "IQCELL: A platform for predicting the effect of gene perturbations on developmental trajectories using single-cell RNAseq data" for consideration at PLOS Computational Biology.

As with all papers reviewed by the journal, your manuscript was reviewed by members of the editorial board and by several independent reviewers. In light of the reviews (below this email), we would like to invite the resubmission of a significantly-revised version that takes into account the reviewers' comments.

We cannot make any decision about publication until we have seen the revised manuscript and your response to the reviewers' comments. Your revised manuscript is also likely to be sent to reviewers for further evaluation.

Sincerely,

Qing Nie

Associate Editor

PLOS Computational Biology

Jason Papin

Editor-in-Chief

PLOS Computational Biology

Reviewer's Responses to Questions

**Comments to the Authors:**

Reviewer #1: Overview:

In this manuscript, Heydari et al. present a new software tool IQCell for inferring and studying gene regulatory networks (GRNs) in scRNA-seq datasets, with a specific focus on applications in stem cells and developmental processes. A key innovation of this tool is that it provides a comprehensive platform for users to infer GRNs directly from scRNA-seq data and then also perform various analyses and simulations. It appears to be assembled from many pre-existing components in a way that is highly useful for the field of scRNA-seq analysis, and includes novel aspects such as segmentation of pseudotime for GRN analysis, a filter for transcription factors to facilitate GRN construction, and GRN model building/testing.

The authors demonstrate that their method can effectively recapitulate known GRN structures and behaviors in their test case of T-cell development. This analysis of the T-cell scRNA-seq dataset from (Zhou et al. 2019) is quite extensive, including comparison of GRN-predicted interactions to the experimentally validated gene interactions from (Longabaugh et al., 2017), their own newly-generated scRNA-seq analysis of fetal liver hematopoietic progenitor cells, and comprehensive in silico simulation of gene-level perturbations, performed with their IQCell-derived GRNs. In my opinion, the insight provided by these analyses is largely corroboratory to previous studies, and is used to demonstrate the utility of the IQCell software tool rather than presenting new insight into T cell development in this manuscript. For this reason, I request that the authors and editors consider whether this manuscript would be better served as a Software Submission rather than a Research Article for PLOS Computational Biology. Perhaps this is a minor bookkeeping concern, but I think it’s worth thinking about, because the Software Submission has advantages for this type of paper.

Along the same lines, if the authors are focused on demonstrating the general utility of the IQCell software tool in this manuscript, then they should test this on more systems rather than just T cell development. I’m generally supportive of this manuscript for publication in PLOS Computational Biology, but if my reading of the authors’ intended focus is correct, then I can’t support publication without validation in additional test systems. My general and specific comments are largely related to this point. If the authors demonstrate that their method can be applied effectively beyond this one well-characterized T cell example application, I will enthusiastically support its publication.

General Comments:

1. The authors claim that their method, “is generally applicable to scRNA-seq data derived from dynamic developmental systems and should serve as a useful resource for many applications,” and “is not specific to a single developmental system ... may be broadly useful in understanding how GRNs contribute to cell development in a variety of developmental contexts”. However, they have not provided any results from testing their method outside of the context of T-cell development. Can you demonstrate that this pipeline was not essentially overfitted to the specific T-cell developmental context? I assume the ideal utility of this method would be in applying it to under-studied applications to discover GRNs de novo… could this actually be applied where we have not yet mapped out GRNs to assist in manually checking and curating the method outcomes, or where GRNs have different and potentially more complex structures? Analysis of one additional system in addition to T cell development should be sufficient, although more would be very welcome.

2. How frequently can you find scRNAseq datasets that meet the specified quality guidelines? Was this tested on other datasets beyond the two presented in this manuscript to establish these quality criteria? This again gets at how widely applicable this method can actually be.

Specific Comments:

1. Please discuss whether the algorithmic decisions, such as using pySCENIC and MAGIC, etc. were based on testing outcomes against alternatives methods? And were these tested with multiple datasets?

2. It would be helpful to provide succinct guidance for users selecting hyperparameters (e.g. k-means clustering or mean-based cutoff for gene expression binarization), perhaps included with the python module documentation.

Reviewer #2: The manuscript presents an interesting computational pipeline to reconstruct and simulate Boolean gene regulatory networks from scRNA-seq data. The model requires count matrix, pseudotime ordering and a curated list of genes (which is optional) and allows to simulate temporal dynamics as well as exploring the effect of perturbations, such as knock outs and overexpressions. The main weakness of the method is that it has been only applied to one experimental system (T-cell development), therefore making it difficult to prove its generality. I have listed below more specific comments about the model’s formulation and application to the T-cell dataset.

Questions on Model:

Fig. S1B does not really explain how the MAGIC method deals with dropout genes. While I understand that this an already existing method, the main steps of MAGIC should be discussed. Since it is an existing method, adding as supplementary could suffice.

The criteria for gene selection are not very clear, it is only stated that biological information and variation along pseudotime were considered. Was the list curated based on literature, and then refined using the gene dynamics score, etc..?

It would be useful to visualize this gene dynamics score for some selected genes, and examples of genes that did not make the cut based on this variation principle.

How do the final networks scale as the threshold to keep the GRN connections is changed?

In the methods, it is mentioned that users can optionally include self-regulation, but no more information is provided. However, treating self-regulation seems a fundamentally different problem because IQCELL uses gene-gene correlations to infer the network connections. Please explain.

Also, IQCELL discretizes genes expression between Low and High during GRN inference. It would be interesting to see how well some of these genes naturally separate between two ON and OFF states. This is a central assumption in building the Boolean network, since more continuous distributions of gene expression cannot be captured.

What is the rationale for allowing up to 4 activators and up to 2 inhibitors in the GRN? How does the predicted network change when these numbers are varied (perhaps include some examples as supplementary figures)?

Also, assuming that all activators must be ON to achieve activation seems a very strict assumption, especially in situations where TFs compete over the same binding sites, and thus not all of them are necessary to start transcription. Is there any evidence for this assumption for the gene lists curated in the manuscript? This could be a case where users should have the possibility to modify these assumptions. Also, it could be interesting to see if relaxing these requirements would fundamentally change the topology of the inferred networks.

Analysis of T-cell:

It is quite difficult to catch the difference between Fig. 2D and 3A, I suggest some alternative way to plot to highlight the differences.

A color shading to visualize time progression would help to characterize the model’s transition states in all PC1-PC2 scatter plots

The fact that automatic selection can identify many of the curated genes is a very strong finding. However, it does not justify claiming that automatic feature selection can robustly replace curated lists, as the authors had to include some genes to reproduce their results previously obtained with curated lists.

**Have the authors made all data and (if applicable) computational code underlying the findings in their manuscript fully available?**

Reviewer #1: Yes

Reviewer #2: Yes

PLOS authors have the option to publish the peer review history of their article (what does this mean?). If published, this will include your full peer review and any attached files.

Reviewer #1: **Yes: **Eli R. Zunder

Reviewer #2: **Yes: **Federico Bocci
---

## [Decision Letter · Decision Letter 1]

10 Jan 2022

Dear Prof. Zandstra,

Thank you very much for submitting your manuscript "IQCELL: A platform for predicting the effect of gene perturbations on developmental trajectories using single-cell RNAseq data" for consideration at PLOS Computational Biology. As with all papers reviewed by the journal, your manuscript was reviewed by members of the editorial board and by several independent reviewers. The reviewers appreciated the attention to an important topic. Based on the reviews, we are likely to accept this manuscript for publication, providing that you modify the manuscript according to the review recommendations.

Sincerely,

Qing Nie

Associate Editor

PLOS Computational Biology

Jason Papin

Editor-in-Chief

PLOS Computational Biology

[LINK]

Reviewer's Responses to Questions

**Comments to the Authors:**

Reviewer #2: The authors have largely addressed my comments. The manuscript reads well and I am generally in favor of acceptance. My only further suggestion is to test the method on some in silico GRNs in order to compare the results with other existing methods for GRN inference. There are many ODE or boolean models of transcription factor networks that could be used to generate in silico data. One (relatively) quick way to compare to many existing methods is Beeline (https://www.nature.com/articles/s41592-019-0690-6). I think this will make the paper much more interesting to people that could actually use the method.

Reviewer #3: The authors have adequately addressed the questions raised by the reviewer #1.

In the current manuscript, the pseudo-time ordering is inferred by DDRtree, which is not the popular one among many pseudo-time inference methods. I wonder whether other pseudo-time methods would affect the result? Also, the applications seem only for single trajectory. How to apply the tool to the branching trajectories should be considered or discussed at least.

In addition, I have some suggestions as follows:

1. Now in the manuscript the network edges were all visualized as arrows for both positive and negative regulations. However, it’s better to distinguish them using shear head arrows and round head arrow, as most GRNs commonly appear.

2. Although the figures are in 300 dpi resolution, but some fonts in the figures are not very clear. The authors may need to improve them.

3. The manuscript should be carefully proofread and edited. Some typos (e.g.,): "Transcription factor" should be consistently "transcription factor"; Line 140.

**Have the authors made all data and (if applicable) computational code underlying the findings in their manuscript fully available?**

Reviewer #2: Yes

Reviewer #3: None

PLOS authors have the option to publish the peer review history of their article (what does this mean?). If published, this will include your full peer review and any attached files.

Reviewer #2: **Yes: **Federico Bocci

Reviewer #3: No

Figure Files:

Data Requirements:

Reproducibility:

References:

---

## [Editor Report · Decision Letter 2]

8 Feb 2022

Dear Prof. Zandstra,

We are pleased to inform you that your manuscript 'IQCELL: A platform for predicting the effect of gene perturbations on developmental trajectories using single-cell RNAseq data' has been provisionally accepted for publication in PLOS Computational Biology.

Best regards,

Qing Nie

Associate Editor

PLOS Computational Biology

Jason Papin

Editor-in-Chief

PLOS Computational Biology

---

## [Editor Report · Acceptance letter]

22 Feb 2022

PCOMPBIOL-D-21-01540R2 

IQCELL: A platform for predicting the effect of gene perturbations on developmental trajectories using single-cell RNAseq data

Dear Dr Zandstra,

I am pleased to inform you that your manuscript has been formally accepted for publication in PLOS Computational Biology. Your manuscript is now with our production department and you will be notified of the publication date in due course.

With kind regards,

Zsofia Freund
